# A Methodological Review of Mixed Methods Research in Palliative and End-of-Life Care (2014–2019)

**DOI:** 10.3390/ijerph17113853

**Published:** 2020-05-29

**Authors:** Sergi Fàbregues, Quan Nha Hong, Elsa Lucia Escalante-Barrios, Timothy C. Guetterman, Julio Meneses, Michael D. Fetters

**Affiliations:** 1Department of Psychology and Education, Universitat Oberta de Catalunya, 08018 Barcelona, Spain; jmenesesn@uoc.edu; 2Evidence for Policy and Practice Information and Co-ordinating Centre (EPPI-Centre), University College London, London WC1H 0NR, UK; quan.nha.hong@mail.mcgill.ca; 3Department of Education, Universidad del Norte, Barranquilla 0000, Colombia; eescalante@uninorte.edu.co; 4Graduate School of Health, Creighton University, Omaha, NE 68178, USA; tguetter@med.umich.edu; 5Department of Family Medicine, University of Michigan, Ann Arbor, MI 48104, USA; mfetters@med.umich.edu

**Keywords:** palliative care research, end-of-life research, mixed methods research, qualitative research, quantitative research, research design, reporting quality

## Abstract

Mixed methods research has been increasingly recognized as a useful approach for describing and explaining complex issues in palliative care and end-of-life research. However, little is known about the use of this methodology in the field and the ways in which mixed methods studies have been reported. The purpose of this methodological review was to examine the characteristics, methodological features and reporting quality of mixed methods articles published in palliative care research. The authors screened all articles published in eight journals specialized in palliative care between January 2014 and April 2019. Those that reported a mixed methods study (n = 159) were included. The Good Reporting of a Mixed Methods Study (GRAMMS) criteria were used to assess reporting quality. Findings showed that 57.9% of the identified studies used a convergent design and 82.4% mentioned complementarity as their main purpose for using a mixed methods approach. The reporting quality of the articles generally showed a need for improvement as authors usually did not describe the type of mixed methods design used and provided little detail on the integration of quantitative and qualitative methods. Based on the findings, recommendations are made to improve the quality of reporting of mixed methods articles in palliative care.

## 1. Introduction

In the last few years, researchers have increasingly recognized mixed methods research as a valuable approach that can enhance the evidence base in palliative care and end-of-life research [1,2]. Mixed methods research relies on a set of designs and procedures that involve the integrated use of qualitative and quantitative methods in a single study or sustained program of inquiry [3]. By combining the strengths of these two families of methods to generate a whole that is greater than the sum of its parts [4], researchers can better address the multi-faceted nature of the phenomena investigated in palliative care. Mixed methods inquiry can help researchers gain a more comprehensive understanding of the health status of patients at the end of their lives by quantitatively generating an account of their symptoms as well as qualitatively capturing their individual experiences of chronic illness [5].

Mixed methods research can also play a crucial role in describing and explaining the complexity and challenges involved in integrating palliative care into healthcare systems. Farqhuar and colleagues [1,5] argue that, since mixed methods studies can provide answers to a wide range of research questions, they are particularly useful in developing and evaluating complex interventions such as symptom management strategies, end-of-life care delivery models, and psychosocial and spiritual services. By highlighting the importance of the context of the intervention under study, mixed methods research can generate evidence not only of the effectiveness of the intervention, but also on how it was delivered in practice and perceived by patients and families [2]. The usefulness of combining qualitative and quantitative methods in intervention research was reinforced in the MORECare Consensus Exercise, published in 2013 [5]. In this study, 33 delegates agreed on a set of recommendations for designing and implementing mixed methods intervention studies to address the key challenges posed by palliative care and end-of-life research.

Two reviews examining the use of mixed methods designs in palliative care have been published to date. In 2008, Flemming, Adamson, and Atkin [6] published a review of 146 trials included in six Cochrane systematic reviews and found only one that incorporated a qualitative component. Later, using a systematic search of articles published between 2010 and 2012, Seymour [7] found 28 mixed methods studies on topics related to supportive and palliative care. Her findings led to the conclusion that “mixed methods studies are becoming more frequently employed in palliative care research and resonate with the complexity of the palliative care endeavour”. However, two limitations apply to the current state of knowledge of the use of mixed methods research in palliative care. First, in the seven years since the last review was undertaken, a considerable number of new mixed methods studies may have been published (coinciding with the considerable increase in the application of mixed methods research in health sciences, as noted by Kaur and colleagues [8]). Second, neither of the two reviews mentioned above examined the reporting quality of the studies under review to determine the extent to which they provide sufficient and transparent information on the design, data collection, analysis and integration procedures used [9]. Complete and clear reporting of all of these methodological aspects is essential to ensure that the evidence generated is robust, relevant and transferable to policy and clinical practice in palliative care. Therefore, the aim of this methodological review is to examine how mixed methods research has been used and reported in the articles published in eight palliative care journals between 2014 and 2019. The following specific aims were addressed: (1) to describe the characteristics of the mixed methods articles published in these journals; (2) to examine the reporting quality and the mixed methods features of these articles; and (3) to examine the differences in reporting quality across the types of mixed methods designs used.

## 2. Materials and Methods

### 2.1. Search Strategy

In this review, we used a journal-focused search strategy similar to that used by Wisdom and colleagues [10] and Bishop and Holmes [11] in two reviews that appraised the quality of mixed methods articles in the fields of health services research and complementary and alternative medicine, respectively. Specifically, we examined all the articles published between January 2014 and April 2019 in the following eight journals: *Palliative Medicine*, *Journal of Palliative Medicine*, *BMJ Supportive & Palliative Care*, *BMC Palliative Care*, *American Journal of Hospice & Palliative Medicine*, *Journal of Palliative Care*, *Journal of Hospice & Palliative Nursing*, and *Palliative & Supportive Care*. These journals were selected because they are specialized in palliative care, and therefore, all the articles published in them are substantively relevant; they publish empirical articles; they are international in scope; and they are respected and well established in the field (they are indexed in the Journal Citation Reports and listed in the websites of the International Association for Hospice and Palliative Care and the CareSearch Palliative Care Knowledge Network). The search was carried out on 28 April 2019.

### 2.2. Study Selection

The titles and the abstracts of the articles published in the eight abovementioned journals during the five-year range examined were downloaded from the PubMed database (i.e., using the journal and publication date search fields) and imported into EPPI-Reviewer software version 4, which was used to manage the study selection. Two independent reviewers (S.F. and Q.N.H.) screened a random sample of 20% of the articles. Disagreements were resolved through discussion with recourse to a third reviewer, if necessary. One reviewer (S.F.) screened the remaining articles. In the next step, the full text of all the articles considered eligible was retrieved and independently assessed by the same two reviewers (S.F. and Q.N.H.). Disagreements at this stage were again resolved by consensus. Reasons for the exclusion of full-text articles were documented.

### 2.3. Screening and Eligibility Criteria

In order to be included in the review, articles needed to report an empirical study involving the collection of quantitative and qualitative data and the use of quantitative and qualitative analyses. They also had to meet the following criteria: provide evidence of integration of the qualitative and quantitative components; include a description of where and how the integration was carried out; refer to an attempt at integrating methods or else use words associated with integration [12]. The authors excluded articles reporting systematic reviews and non-empirical articles, including protocols, theoretical and methodological papers, editorials, commentaries, letters to the Editor, and book reviews. The following types of empirical articles were also excluded: (1) articles reporting the use of either quantitative or qualitative research alone, (2) articles reporting only the quantitative component or the qualitative component of mixed methods studies, (3) single-method studies based on either surveys with open and closed-ended questions or interviews complemented by a supplementary quantitative instrument, and (4) articles in which the information from only one qualitative data source was analyzed quantitatively or only one quantitative source was analyzed qualitatively. While some authors could argue that categories 3 and 4 could be considered mixed methods studies, we excluded these due to the lack of rigor relative to the mixed methods tradition and, in the case of category 4, because they also fail to leverage the full strength of the qualitative methodology.

### 2.4. Data Extraction and Coding

A standardized form was created in an Excel spreadsheet to extract data from the articles included (see Appendix A). The following information was extracted: publication metadata (i.e., publication year, geographical area of the corresponding author, journal name), study purpose, procedures followed in the qualitative and quantitative components (i.e., sampling, data collection, analysis), and features characterizing the mixed methods component (i.e., justification for using mixed methods, type of mixed methods design used, evidence of integration). Similar to methods used in previous reviews appraising the quality of mixed methods studies in the health sciences [10], data extraction was performed by one reviewer (S.F.), and a second reviewer (Q.N.H.) double-checked a random sample (20% of the total sample) for accuracy and consistency. Once extraction was completed, the information was coded to facilitate data synthesis and analysis. A coding scheme was developed that was informed by the methodological literature on mixed methods research, including the Good Reporting of a Mixed Methods Study (GRAMMS) criteria proposed by O’Cathain, Murphy, and Nicholl [13] (see the complete coding scheme in Appendix A). To ensure consistency and avoid drift in carrying out the coding, the same two independent reviewers (S.F. and Q.N.H.) applied the coding scheme to another random sample (20%) of articles. As in the screening and eligibility phases, disagreements between the reviewers on the interpretation and application of the codes were resolved by discussion, with the involvement of a third reviewer. One reviewer (S.F.) coded the remaining articles. Descriptive statistics and crosstabs were used to analyze the coded information. Since the sample size of articles included in the review was relatively small, and for several cells the expected count was less than five, the standard Pearson’s chi-square test was inappropriate. Therefore, Fisher’s exact test was used to determine the statistical significance of differences between groups. Qualitative content analysis [14] of the data extracted from the articles was also carried out in order to identify patterns in the methodological components of the studies reviewed and to complement the quantitative findings.

## 3. Results

The initial search generated a total of 5136 articles. After assessing eligibility, we included 159 articles reporting empirical mixed methods studies (see complete list of articles in Appendix A). Figure 1 shows the Preferred Reporting Items for Systematic Reviews and Meta-Analyses (PRISMA) flowchart of the review.

### 3.1. Characteristics of Articles Reporting Mixed Methods Studies

As shown in Table 1, of the 5136 articles published between January 2014 and April 2019 in the eight palliative care journals that were identified, 3225 were empirical articles. Of these, less than 5% (n = 159) met the study’s criteria for mixed methods studies. The journals with the highest frequency of mixed methods studies as a percentage of total articles were BMC Palliative Care (8.2%), Palliative Medicine (7.0%), BMJ Supportive & Palliative Care (5.4%), and Palliative & Supportive Care (5.4%).

Table 2 shows the characteristics of the 159 articles included in this review. In most cases, the corresponding author was affiliated with a university or institution located in Europe (45.3%) or North America (41.5%). In half of the articles from Europe, the corresponding author was affiliated in the United Kingdom. To describe mixed methods, 67.9% of the articles used either the term ‘mixed methods’ (n = 103) or related terms such as ‘multi-method’ (n = 3) and ‘multiple methods’ (n = 2), while 32.1% did not use any of these terms. Fewer than one-quarter of articles (23.9%) cited at least one methodological reference to mixed methods research in order to provide a justification for using this type of inquiry, the most frequently cited being the works authored by Prof. John W. Creswell. Only one of these references was specific to palliative care [1] (see the most frequently cited literature in Appendix A).

We used a typology of study topics from a recent review of palliative care research [15] to categorize the substantive content of the articles. More than a third (39.0%) of the topics researched were related to the planning of care, the transition from curative to palliative care services, and decision-making processes relating to the type and place of care. Around a quarter of the articles (26.4%) focused on the organizational and professional development of palliative care, including aspects such as the development of curricula to teach palliative care and the use of registers for measuring the quality of end-of-life care. Other topics studied include the existential and ethical dimensions of palliative care and the patients’ and family members’ experiences with illness.

Nearly half of the articles (43.4%) evaluated an intervention, program, or service, such as a mindfulness-based stress reduction training program for lung cancer patients and their partners [16], a conversation game to engage individuals in end-of-life discussions [17], and a home-based palliative care service [18]. Other study purposes included investigating a research topic in palliative care and end-of-life research (32.1%), developing a tool (11.9%), a quantitative instrument (6.3%) or an intervention (3.8%), and assessing palliative care needs (2.5%) (see the specific focus of each article in Appendix A).

### 3.2. Reporting Quality and Mixed Methods Features

None of the articles included in the review fully conformed with all six GRAMMS criteria for good reporting of mixed methods studies. Six articles (3.8%) fulfilled 5 criteria, 16 articles (10.1%) fulfilled 4 criteria, and 137 articles (84.9%) fulfilled only 3 criteria or less. Table 3 classifies the reporting quality of the 159 articles in terms of their compliance with each of the six GRAMMS criteria.

Table 4 provides further detail on the mixed methods features of these articles.

#### 3.2.1. Justification for Using Mixed Methods Research

More than half of the articles (62.9%) explicitly justified the use of mixed methods research. One-third (30.2%) provided no explicit justification, but the justification could still be inferred from the objectives of the quantitative and qualitative components. A few articles (6.9%) neither provided a justification nor included the information needed to infer why mixed methods were being used.

Among the rationales for mixing methods, complementarity was the most prevalent (82.4%), including arguments such as the following: to gain a more comprehensive and holistic understanding of the investigated phenomena, to enhance the findings of one component with the findings of the other component, and to answer different types of questions in the testing and evaluation of complex interventions. For example, Reese and Beckwith [22] conducted an online survey with hospice directors to identify barriers to the development of culturally competent hospice care programs and then collected additional qualitative data on strategies being used to overcome these barriers. In the articles reporting evaluations of interventions, the complementarity purpose was reflected in the use of quantitative data to examine the feasibility and efficacy of the intervention and in the utilization of qualitative data to better understand elements related to the context and process of its implementation. For instance, in an evaluation of a patient-reported outcome intervention in chronic heart failure, Kane and colleagues [23] gathered quantitative information on intervention recruitment, adherence, and outcomes and, in parallel, also carried out interviews to explore the views of nurses and patients on the acceptability and delivery of the intervention.

The second most prevalent rationale for using mixed methods was development, which was mentioned in 44.6% of the articles that provided a justification for mixing methods. Studies mentioning this type of rationale used the findings of one component to inform the data collection procedures (28.4%) and/or the sampling (16.2%) of the other component. For example, Myers and colleagues [24] used the findings from semistructured interviews and focus groups with patients and clinicians to generate a Chronic Cancer Experiences Questionnaire (CCEQ), which was psychometrically validated with cancer units in the second phase of the study. Finally, triangulation as a rationale was mentioned in only 13.5% of the articles. In these cases, the motives adduced for using mixed methods included the following: to assess the congruence between the quantitative and qualitative data, to validate one type of data, and to improve the legitimacy of the findings.

#### 3.2.2. Mixed Methods Design

Overall, authors poorly reported the types of mixed methods design used. Very few articles (5.0%) explicitly named the mixed methods design used and described the priority (i.e., the weight) and timing (i.e., the order in which they are carried out) of the quantitative and qualitative components. Slightly more than a quarter (27.7%) named the design used but did not include complete information on both the priority and timing of the components. More than two thirds (67.3%) did not report the name of the design. However, all the articles included enough information in their methods section to allow us to categorize the type of mixed methods design used according to the typology of Creswell and Plano Clark [20].

The majority of the articles (57.9%) used a convergent design, which entailed the independent collection of quantitative and qualitative data, the separate analysis of both types of data, and the final merging of the two datasets to see whether the findings converged, diverged or enhanced each other. For example, in a parallel convergent mixed methods study on clinical decision-making at the end of life, Taylor and colleagues [25] compared the findings from two independent quantitative and qualitative datasets in order to identify “areas of agreement, partial agreement, silence and dissonance”. This comparison allowed them to reinforce findings that were common to the two datasets, including the conclusions about the time-dependent nature and inherent uncertainty of clinical decision-making processes in palliative care.

The other designs used in the reviewed articles were the exploratory sequential design (the quantitative component builds on the qualitative component carried out in the first phase) in 10.1% of the cases, the explanatory sequential design (the qualitative component builds on and/or helps to explain the quantitative component carried out in the first phase) in 18.9% of the cases, and the multistage design (multiple stages are carried out, with any combination of convergent and sequential designs) in 13.2% of the cases.

We examined the relationship between type of design and study purpose since these two elements are closely associated, as frequently stated in the mixed methods literature [20,26]. Table 5 shows a significant relationship (*p* < 0.001) between the study purpose and the type of mixed methods used: while convergent and explanatory sequential designs were used mainly for evaluating interventions or investigating research topics in palliative care, exploratory sequential, and multistage designs were generally employed to develop a tool, a quantitative instrument, or an intervention.

#### 3.2.3. Quantitative and Qualitative Components

Both the quantitative and qualitative methods used were, in general, appropriately reported individually. Over half of the articles (66.7%) provided an adequate and complete description of the procedures followed in the sampling, data collection, and analysis stages of both components, while almost a third (32.1%) provided an almost adequate and complete description of these stages. The reporting of the quantitative and qualitative methods used was insufficient in only two articles. In the articles with limited reporting, it was often the qualitative data analysis stage that displayed significantly poorer reporting. In these cases, crucial elements were missing or insufficiently described, for example, the type of qualitative data analysis approach used and the procedures followed in the development of the coding scheme.

#### 3.2.4. Integration

Two thirds of the articles (66.7%) reported specific evidence of integration. Some of the articles (10.7%) did not include evidence of integration, but they at least described where and how integration had occurred, while almost one-quarter of the articles (22.6%) included no explicit description of integration or evidence of its use. In the subset of articles that reported evidence and/or provided a description of integration, we coded the ways in which integration was carried out at the methods level, using Fetters, Curry, and Creswell’s [21] typology of integration practices. The integration method used in most of the articles (82.1%) was merging (bringing together the quantitative and qualitative findings or data for comparison or analysis). Over one third (35%) integrated through building (using the findings of one component to help build the data collection instruments of the other component) and only a few (17.9%) integrated through connecting (using the findings of one component to inform the sampling strategy followed in the other component).

Using the same typology [21], we coded the ways in which integration was reported. Of the articles that reported evidence of integration, the vast majority (82.1%) integrated through narrative, a few (14.2%) integrated through joint display, and a very small number (3.8%) integrated through data transformation. In the articles integrating through narrative, the authors generally explained verbally whether the relationship between the quantitative and the qualitative findings was one of confirmation, expansion, or discordance [21]. For instance, in their explanatory sequential mixed methods study, Resse and Beckwith [22] provide an example of discordance when they state that many of the barriers to developing hospice care programs that were cited in the qualitative data as having major importance were, in contrast, the ones that were rated of lowest importance in the quantitative data. In the articles integrating through joint displays, the linking between the findings of both components was visually represented in the form of a table, diagram, or matrix whose main function was to highlight the additional insights generated from using a mixed methods design. For example, a joint display table was included in the convergent mixed methods study by Taylor and colleagues [25]. By arranging in the same row the quantitative and qualitative findings that converged and those that diverged and by assigning a different color to each row, the authors were able to more efficiently compare the two types of data and provide a better contextualization of the findings from each dataset. Finally, in the articles integrating through data transformation, the qualitative data were transformed into counts and then integrated with a quantitative database. For instance, in a study [27] examining the relationship between the delivery of psychosocial care by nurses and their familiarity with patients, the authors converted a set of qualitative data on the ways in which nurses responded to the psychosocial needs of the patients into quantitative variables. Then, the authors cross-tabulated these newly quantitative variables with other variables representing the individual characteristics of the nurses.

#### 3.2.5. Limitations and Insights

Only 3.8% of the articles reported specific limitations arising from the use of “one method associated with the presence of the other method” [13]. An example of this type of limitation was found in the explanatory sequential mixed methods study carried out by Boss and colleagues [28]. In this study, the authors recognized that the views concerning anxiety management that the nurses expressed in the (second) qualitative stage of the study may have been influenced by the awareness of this topic that they developed while carrying out the (first) quantitative stage of the study.

Only slightly more than a quarter of the articles (26.4%) described the insights gained from integrating methods. These descriptions complemented the authors’ previously stated justification for using mixed methods by highlighting the ways in which integration helped to “provide greater insight into the feasibility, acceptability and implementation” [23] of a complex intervention, to “reveal a deeper understanding of the way things work [in an intervention] and thereby facilitate transferability to other settings” [29], to “provide valuable, multi-faceted outcome data over time” [30], to “strengthen the validity of the results achieved” [31], and to “facilitate application of theory and research into clinical practice” [32].

### 3.3. Differences in Reporting Quality across Types of Mixed Methods Designs

According to Creswell and Plano Clark [20], mixed methods designs involve particular procedures and decisions that guide the ways in which they are reported. Therefore, since each type of design involves a specific form of reporting, we examined the differences in the reporting quality of the articles according to the type of mixed methods design used (Table 6).

Table 6 shows a significant relationship (*p* < 0.001) between the type of mixed methods design used in the articles and two GRAMMS reporting criteria: the criteria on providing a justification for using mixed methods research (criterion 1) and on describing and presenting evidence of integration (criterion 4). With respect to these two criteria, Fisher’s exact test rejected the null hypothesis that mixed methods research designs are equally likely to show the same reporting quality. Specifically, in both criteria the reporting quality was poorer in the studies using a convergent design than in those using sequential or multistage designs (i.e., the justification for using mixed methods and the integration process were both less frequently described in convergent mixed methods studies).

## 4. Discussion

### 4.1. Main Findings

This review has described the characteristics, methodological features, and reporting quality of the mixed methods articles published in eight palliative care journals between 2014 and 2019. Our findings show that fewer than 5% of the empirical articles published during the six-year period under study used a mixed methods design. While this low frequency of articles is consistent with the results of previous reviews on mixed methods research in the health sciences [10,11,33], it belies the unanimous agreement on the usefulness of this methodological approach in answering the kind of complex questions that are being asked in palliative care and end-of-life research [1].

In line with findings reported in recent reviews [8,26,34], many articles reported a convergent design and cited complementarity as the main rationale for mixing methods. In these studies, the findings of the quantitative and the qualitative components were used to enrich or elaborate on the findings of the other component or to provide further insights. Many of these articles reported evaluation studies in which quantitative and qualitative data were used to holistically assess the different components of an intervention while incorporating contextual information provided by health professionals, patients, and families. The high proportion of mixed methods evaluation studies relative to non-evaluation studies found in our review clearly demonstrates that palliative care researchers are aware of the value of mixed methods studies in providing a more comprehensive evaluation of complex health care interventions, as argued by Farqhuar and colleagues [1].

A key finding of this review is that the reporting of the mixed methods articles published in the eight journals examined showed a need for improvement. In line with previous reviews [8,35], none of the articles included in our review fulfilled all six GRAMMS criteria and none of the six criteria was fulfilled by all the articles. The reporting quality was not consistent across the six criteria, with variations according to the particular mixed methods feature being examined. The two worst reported features were the limitations of using one method associated with the presence of the other method and the insights gained from using mixed methods; in most studies, neither of these features were described. By contrast, the justification for using mixed methods was explicitly described in most articles, although the motive was less well reported when a convergent design was used. In such studies, the rationale had to be inferred, when possible, from the individual objectives of both the quantitative and qualitative components. More problematic was the reporting of the mixed methods design used, since the authors frequently failed to mention the type of design and to provide information on the priority and sequence of the components.

The findings also demonstrate that there is considerable room for improvement in the reporting of the integration of the quantitative and qualitative components of studies. Two important problems were observed. First, one-quarter of the articles failed to report any evidence of integration. Most of these were convergent mixed methods studies that separately presented the quantitative and qualitative findings and discussed them in separate paragraphs in the discussion section. In these studies, the failure to show the outcome of the integration is clearly important since this omission makes it difficult for readers to identify what insights were gained from mixing methods. Arguably, convergent studies are most amenable to integration since both data outcomes are available at the time of interpretation. Second, the vast majority of the articles that provided evidence of mixing used narrative to report the integration, and very few used joint displays. The infrequent use of joint displays in reporting integration is further cause for concern since joint displays are a more efficient way than narrative of facilitating a “direct and nuanced comparison of the [quantitative and qualitative] results” [20] and generating “new inferences” [36].

### 4.2. Implications for Future Research

This review highlights important implications for promoting and facilitating the use of mixed methods research in palliative care and improving the reporting quality of published research. The low frequency of mixed methods articles that we found in eight journals could be explained by the existence of a number of practical barriers to carrying out this type of research. These include the pressure to generate evidence rapidly in dynamic healthcare environments, along with the need to obtain extensive funding, build interdisciplinary teams of qualified researchers with varied methodological skills, and deal with potential disagreements and power differentials within these teams [1,4,37]. Researchers need to be aware of these barriers and should discuss possible ways to overcome them in order to increase the number of mixed methods studies published in palliative care journals.

Our findings also highlighted the need to improve the reporting quality of mixed methods studies in palliative care. Without an accurate description of the methods used in these studies, readers cannot properly assess their methodological quality and replicate the procedures [9,10]. Palliative care and other health researchers face important challenges when reporting mixed methods research. These include the length limitations of journals [26], practitioners’ lack of familiarity with mixed methods research [4], the complexity of reporting integration [21], and authors’ lack of knowledge of reporting guidelines [35].

### 4.3. Recommendations to Researchers

In response to these challenges, a number of recommendations can be made. First, researchers need to make an effort to write concisely in order to be able to represent the complexity of the process and the findings of mixed methods research with sufficient clarity within the length limitations of the journals. Concise and complete presentation of the methods would also make the articles accessible to a wider audience. Second, since integration of methods is an activity that demands specialized methodological skills, researchers should receive specific training in this methodology or else collaborate with researchers and authors knowledgeable in these aspects of research.

### 4.4. Recommendations to Editors

Editors of palliative care journals could play a key role in improving the reporting quality of mixed methods studies by: (1) publishing editorials and methodological articles that include field-specific guidelines for reporting mixed methods studies, (2) encouraging authors and reviewers to use existing published guidelines for reporting mixed methods research [13,38,39] and advising authors on compliance with those guidelines, and (3) publishing well-presented mixed methods studies that can serve as examples of adequate reporting. Adoption of the GRAMMS criteria for reporting mixed methods study findings could lead to a substantial improvement since authors are obliged to meet journal requirements. Table 7 shows four examples of well-reported articles identified in our review that illustrate the ways in which mixed methods can contribute meaningfully to palliative care.

### 4.5. Strengths and Limitations

To our knowledge, this is the first review that examines the reporting quality of mixed methods studies published in palliative care and end-of-life research. A notable feature is how we identified the studies included in the review. By reading the abstracts of all the articles published between 2014 and 2019 in the selected journals, we were able to accurately identify all articles that used mixed methods, including those that did not use this term. A limitation is that only the articles published in eight English language journals were reviewed, so that articles published in other journals were excluded. The search strategy used led to an underestimation of the true prevalence of articles reporting mixed methods studies in the field of palliative care and end-of-life research. Furthermore, we used a narrow definition of mixed methods research that excluded some studies that fall into a grey area. For example, there is a debate in the literature on whether surveys with open and closed-ended questions or studies that transform qualitative data into quantitative data should be considered mixed methods research. In this review, these types of studies were excluded.

## 5. Conclusions

Mixed methods research has wide-ranging applications in the field of palliative care. Our findings have confirmed that mixed methods designs can be especially valuable for addressing the multi-faceted nature of palliative care phenomena, as well as for evaluating complex healthcare interventions. Furthermore, authors are only beginning to familiarize themselves with and use existing guidelines for reporting mixed methods studies. This review makes a series of recommendations for researchers and journals that could enhance the quality of reporting of mixed methods research. More careful attention to the requirements for reporting could help researchers to fulfill the potential of mixed methods research to provide a more comprehensive understanding of the many complex issues that palliative care researchers need to address.

## Figures and Tables

**Figure 1 ijerph-17-03853-f001:**
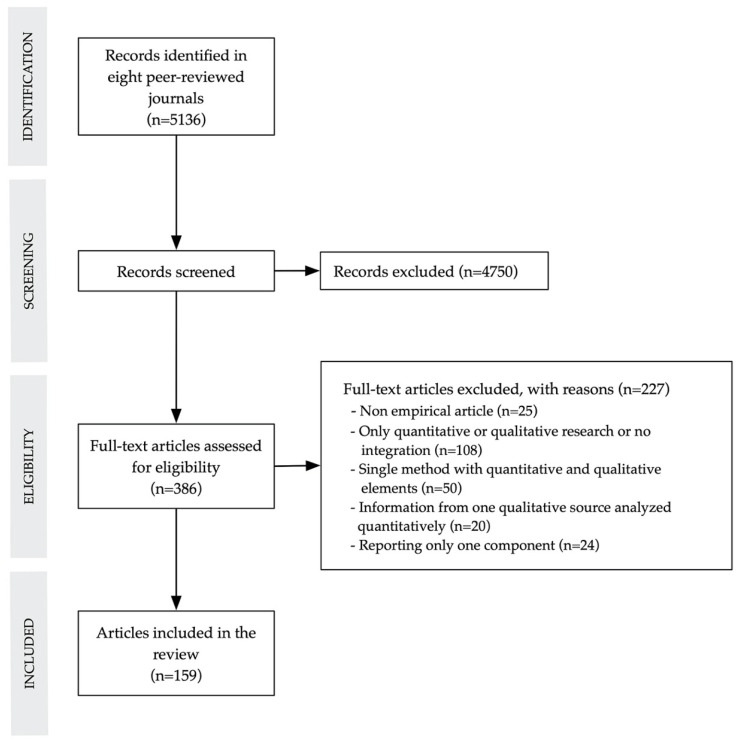
Preferred Reporting Items for Systematic Reviews and Meta-Analyses (PRISMA) flowchart.

**Table 1 ijerph-17-03853-t001:** Frequency of mixed methods studies published in eight palliative care journals between 2014 and 2019.

Journals	Year								
	Jan 2014	2015	2016	2017	2018	April 2019	Total	Mixed Methods Studies	%
American Journal of Hospice & Palliative Medicine	91	102	106	115	186	98	698	26	3.7
BMC Palliative Care	50	56	77	76	97	30	386	32	8.2
BMJ Supportive & Palliative Care	33	53	43	57	77	48	311	17	5.4
Journal of Hospice & Palliative Nursing	37	44	40	45	48	17	231	8	3.4
Journal of Palliative Care	13	22	0	15	39	11	100	3	3.0
Journal of Palliative Medicine	114	115	134	135	181	23	702	23	3.2
Palliative & Supportive Care	51	75	51	53	104	35	369	20	5.4
Palliative Medicine	64	73	63	65	123	40	428	30	7.0
Total	453	540	514	561	855	302	3225	159	4.9

**Table 2 ijerph-17-03853-t002:** Characteristics of the 159 articles included in the review.

Characteristics	n (%)
*Geographical area of the corresponding author*	
Africa	1 (0.6)
Asia	7 (4.4)
Europe	72 (45.3)
North America	66 (41.5)
Oceania	13 (8.2)
*Study identification regarding mixed methods*	
Self-identified as mixed methods	108 (67.9)
Non-identified as mixed methods	51 (32.1)
*Cited key literature on mixed methods*	38 (23.9)
*Study topic*	
Care planning, place of care, transition, and documentation	62 (39.0)
Caring, situations, and relationships	7 (4.4)
Existential and ethical issues	15 (9.4)
Experiences of illness, well-being, needs, and environment	12 (7.5)
Organizational or professional development	42 (26.4)
Symptom assessment and management	10 (6.3)
Other topics	11 (6.9)
*Study purpose*	
Assessment of palliative care needs	4 (2.5)
Evaluation of an intervention in or program or service for palliative care	69 (43.4)
Investigation of a research topic in palliative care	51 (32.1)
Development and evaluation of an intervention in or program or service for palliative care	6 (3.8)
Development and validation of a quantitative instrument for palliative care	10 (6.3)
Development of a tool or model for palliative care	19 (11.9)

**Table 3 ijerph-17-03853-t003:** Reporting quality of the 159 articles included in the review according to the Good Reporting of a Mixed Methods Study (GRAMMS) criteria.

GRAMMS Criteria	Yes ^1^ n (%)	Yes, but n (%)	No n (%)
1. Describes the justification for using mixed methods research to the research question	100 (62.9)	48 (30.2)	11 (6.9)
2. Describes the mixed methods design in terms of the purpose, priority and sequence of methods	8 (5.0)	44 (27.7)	107 (67.3)
3. Describes each method in terms of sampling, data collection and analysis	106 (66.7)	51 (32.1)	2 (1.3)
4. Describes the integration of the quantitative and qualitative components ^2^	106 (66.7)	17 (10.7)	36 (22.6)
5. Describes any limitation of one method associated with the presence of the other method	6 (3.8)	0 (0)	153 (96.2)
6. Describes any insights gained from mixing or integrating methods	42 (26.4)	5 (3.1)	112 (70.4)

^1^ These categories are described in detail in Appendix A. ^2^ For the purpose of this study, the authors modified the wording of this criterion as compared to the original.

**Table 4 ijerph-17-03853-t004:** Mixed methods research features of the 159 articles included in the review.

Mixed Methods Features ^1^	n (%)
*Justification for using mixed methods research criteria* ^2^ *(n = 148)*	
Complementarity	122 (82.4)
Development	66 (44.6)
To inform data collection	42 (28.4)
To inform sampling	24 (16.2)
Triangulation	20 (13.5)
*Type of mixed methods design (n = 159)*	
Convergent	92 (57.9)
Exploratory sequential	16 (10.1)
Explanatory sequential	30 (18.9)
Multistage	21 (13.2)
*Integration at the methods level* ^2^ *(n = 123)*	
Merging	101 (82.1)
Building	43 (35.0)
Connecting	22 (17.9)
*Integration at the reporting level* ^2^ *(n = 106)*	
Narrative	87 (82.1)
Joint display	15 (14.2)
Data transformation	4 (3.8)

^1^ Mixed methods features were coded using the typologies developed by Plano Clark and Ivankova [19] (for the justification for using mixed methods), Creswell and Plano Clark [20] (for the type of mixed methods design), and Fetters, Curry, and Creswell [21] (for the integration at both the methods and reporting levels). ^2^ Categories are not mutually exclusive. The percentages are calculated relative to the number of articles that included information on this feature.

**Table 5 ijerph-17-03853-t005:** Study purpose of the 159 articles included in the review by type of mixed methods design.

Study Purpose	Type of Design				
	Convergent n (%)	Exploratory Sequential n (%)	Explanatory Sequential n (%)	Multistage n (%)	Total n (%)
Assessment of palliative care needs	2 (2.2)	0 (0)	1 (3.3)	1 (4.8)	4 (2.5)
Evaluation of an intervention in or program or service for palliative care	60 (65.2)	0 (0)	6 (20.0)	3 (14.3)	69 (43.4)
Investigation of a research topic in palliative care	26 (28.3)	4 (25)	19 (63.3)	2 (9.5)	51 (32.1)
Development and evaluation of an intervention in or program or service for palliative care	0 (0)	0 (0)	0 (0)	6 (28.6)	6 (3.8)
Development and validation of a quantitative instrument for palliative care	1 (1.1)	7 (43.8)	0 (0)	2 (9.5)	10 (6.3)
Development of a tool or model for palliative care	3 (3.3)	5 (31.3)	4 (13.3)	7 (33.3)	19 (11.9)

Fisher’s value = 97.328, *p* < 0.001.

**Table 6 ijerph-17-03853-t006:** Reporting quality of the 159 articles included in the review by type of mixed methods design.

GRAMMS Criteria	Type of Design						
	Convergent n (%)	Exploratory Sequential n (%)	Explanatory Sequential n (%)	Multistage n (%)	Total n (%)	Fisher’s Value	*p* Value
1. Describes the justification for using mixed methods research to the research question						34.586	0.001
Yes	41 (44.6)	16 (100)	27 (90.0)	16 (76.2)	100 (62.9)		
Yes, but	42 (45.7)	0 (0)	2 (6.7)	4 (19.0)	48 (30.2)		
No	9 (9.8)	0 (0)	1 (3.3)	1 (4.8)	11 (6.9)		
2. Describes the mixed methods research design in terms of the purpose, priority and sequence of methods						7.368	0.232
Yes	8 (8.7)	0 (0)	0 (0)	0 (0)	8 (5.0)		
Yes, but	21 (22.8)	6 (37.5)	12 (40.0)	5 (23.8)	44 (27.7)		
No	63 (68.5)	10 (62.5)	18 (60.0)	16 (76.2)	107 (67.3)		
3. Describes each method in terms of sampling, data collection and analysis						3.128	0.807
Yes	64 (69.6)	10 (62.5)	20 (66.7)	12 (57.1)	106 (66.7)		
Yes, but	26 (28.3)	6 (37.5)	10 (33.3)	9 (42.9)	51 (32.1)		
No	2 (2.2)	0 (0)	0 (0)	0 (0)	2 (1.3)		
4. Describes the integration of the quantitative and qualitative components						22.570	0.001
Yes	58 (63.0)	11 (68.8)	20 (66.7)	17 (81.0)	106 (66.7)		
Yes, but	4 (4.3)	5 (31.3)	6 (20)	2 (9.5)	17 (10.7)		
No	30 (32.6)	0 (0)	4 (13.3)	2 (9.5)	36 (22.6)		
5. Describes any limitation of one method associated with the presence of the other method						6.124	0.062
Yes	2 (2.2)	0 (0)	4 (13.3)	0 (0)	6 (3.8)		
Yes, but	0 (0)	0 (0)	0 (0)	0 (0)	0 (0)		
No	90 (97.8)	16 (100)	26 (86.7)	21 (100)	153 (96.2)		
6. Describes any insights gained from mixing or integrating methods						6.051	0.347
Yes	29 (31.5)	2 (12.5)	8 (26.7)	3 (14.3)	42 (26.4)		
Yes, but	3 (3.3)	1 (6.3)	0 (0)	1 (4.8)	5 (3.1)		
No	60 (65.2)	13 (81.3)	22 (73.3)	17 (81.0)	112 (70.4)		

**Table 7 ijerph-17-03853-t007:** Examples of well-reported mixed methods studies in palliative care.

Authors	Objective	Justification for Using Mixed Methods	Mixed Methods Design	Data Sources	Integration	Insights Gained from Using Mixed Methods
Van Scoy et al. [17]	Evaluate an end-of-life conversation game	Complementarity [p. 595] ^1^	Convergent [pp. 594–595]	Questionnaires on the confidence experienced by participants during the game, and interviews on participants’ views on the game [p. 595]	Merging, by comparing the quantitative and qualitative findings through a joint display table [pp. 595–598]	The convergence between the two types of findings strengthened the conclusions and facilitated a more comprehensive evaluation of the game [p. 599]
Jors et al. [40]	Investigate interactions between cleaning staff and patients	Development to inform data collection/Triangulation [p. 64]	Exploratory sequential [p. 64]	Interviews and focus groups with cleaning staff on patient interaction and coping with death, and questionnaire distributed to cleaning staff [pp. 64–65]	Building, by using the qualitative findings to generate a questionnaire. Merging, by comparing both findings through narrative [pp. 64–67]	The qualitative findings provided a more comprehensive basis for designing the questions included in the quantitative questionnaire [p. 71]
Zweers et al. [41]	Examine nurses’ knowledge, needs and practices when supporting anxious patients	Complementarity/Development to inform sampling [p. 2]	Explanatory sequential [p. 2]	Online survey with nurses caring for anxious patients on their knowledge, needs and practices, and focus groups with nurses who completed the survey [p. 2–3]	Connecting, by using the survey findings to inform the qualitative sampling. Merging, by comparing both findings through narrative [pp. 2–7]	The quantitative findings helped to define the qualitative sample. The qualitative findings provided deeper insight into the quantitative findings on anxiety management practices by nurses [p. 8]
Knighting et al. [42]	Develop an alert thermometer for carers and non-specialist staff	Complementarity/Development to inform data collection/Development to inform sampling [pp. 3–4]	Multistage [p. 3]	Interviews and focus groups with carers for item generation. Delphi survey with carers and health professionals for item selection [pp. 3–9]	Building and connecting, by using the qualitative findings to inform the Delphi content and sampling. Merging through narrative [pp. 3–9]	Mixed methods enabled researchers to sequentially develop and select the items of the alert thermometer. It also allowed incorporation of views and experiences of a wide range of participants [pp. 10–11]

^1^ Page numbers in brackets indicate where this feature was reported in the original article.

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
