# Peer review of "A Methodological Review of Mixed Methods Research in Palliative and End-of-Life Care (2014–2019)"

_ijerph, 2020, doi:10.3390/ijerph17113853_

Round 1

Reviewer 1 Report

It is a privilege to review this wonderful paper. I thoroughly enjoyed reading it!

The topic is highly relevant and your paper adds an important dimension to it, particularly in the context of palliative care.

The paper needs little changes as it is. A few suggestions are made here for your consideration to make it even stronger.

Title felt like a mouthful and took me a while to appreciate it. How does the following sound? "A methodology review of publications in palliative care and EOL research that use mixed methods designs."

Line 21: "However, little is known... " instead of "there is a lack of knowledge." perhaps.

Line 26: "most of the identified studies... ". Suggest stating the exact figures objectively in the abstract.

Line 28: Instead of "mixing methods" which reads a little awkward, why not use "mixed method approaches." 

Line 31: " … quality of reporting studies using mixed method designs in palliative and end-of-life care." rather than reporting mixed method articles.

Line 36: Which century? 21st? Cited article was published 2011.

Line 39: "sustained program of inquiry". What does this mean exactly?

Section 2.1: Is a sample of the search strategy provided in the supplementary material? Which databases were they run in? Just PubMed? From the findings reported in section 3.1, it seems that each journal was hand searched over a 5 year period instead. This was unclear for me as a reader.

Line 81: "we used the same search strategy". Exactly the same? Or adapted.

Line 117: Did you mean the individual or mixed method "tradition"?

Line 129: What is the purpose of this scheme, particularly after a very structured data extraction? For counting it seems, after reading the last sentence of this para. To explain a little more here. An illustration of an example or the final coding scheme would be most helpful.

Table 2 & 4: The subheadings inside the table could be italized for clarity.

Table 4: Was the categorization informed by a particular methodological paper? May be good to cite it or mention it here (Cresswell & Clark for example mentioned only later in the article).

Same point highlighted in line 418-420 could explain why so little reports of mixed method studies are found in this review (commented in line 337). This may be added explicitly here to illustrate your point as a study limitation.

Wishing your team well and looking forward to seeing your article in print soon.

Author Response

Comment 1: Title felt like a mouthful and took me a while to appreciate it. How does the following sound? "A methodology review of publications in palliative care and EOL research that use mixed methods designs."

Answer: Thanks for your suggestion. We have changed the title and now it reads “A methodological review of mixed methods research in palliative and end-of-life care (2014-2019)”.

Comment 2: Line 21: "However, little is known... " instead of "there is a lack of knowledge." perhaps.

Answer: Done.

Comment 3: Line 26: "most of the identified studies... ". Suggest stating the exact figures objectively in the abstract.

Answer: Done, we have changed the text. Now it reads “Findings showed that 57.9% of the identified studies used a convergent design and 82.4% mentioned complementarity as…”

Comment 4: Line 28: Instead of "mixing methods" which reads a little awkward, why not use "mixed method approaches."

Answer: Done, we have changed the text. Now it reads “…as their main purpose for using a mixed methods approach”.

Comment 5: Line 31: " … quality of reporting of studies using mixed method designs in palliative and end-of-life care." rather than reporting mixed method articles.

Answer: We prefer no to make this change since the term “mixed methods articles” is widely used in the literature.

Comment 6: Line 36: Which century? 21st? Cited article was published 2011.

Answer: Done, we have changed the text. Now it reads “In the last few years…”.

Comment 7: Line 39: "sustained program of inquiry". What does this mean exactly?

Answer: It means a program of studies rather than a single study. We clarified this and now the text reads “in a single study or sustained program of inquiry”.

Comment 8: Section 2.1: Is a sample of the search strategy provided in the supplementary material? Which databases were they run in? Just PubMed? From the findings reported in section 3.1, it seems that each journal was hand searched over a 5 year period instead. This was unclear for me as a reader.

Answer: We agree with the reviewer that this information was not sufficiently explicit. We clarified this and now the text reads “The titles and the abstracts of the articles published in the eight abovementioned journals during the five-year range examined were downloaded from the PubMed database (i.e., using the journal and publication date search fields) and imported into…”.

Comment 9: Line 81: "we used the same search strategy". Exactly the same? Or adapted.

Answer: It is similar, but not exactly the same. We clarified this and now the text reads “In this review, we used a journal-focused search strategy similar to that used by Wisdom and colleagues and Bishop and Holmes…”

Comment 10: Line 117: Did you mean the individual or mixed method "tradition"?

Answer: It is the mixed methods tradition. We have clarified this in the text.

Comment 11: Line 129: What is the purpose of this scheme, particularly after a very structured data extraction? For counting it seems, after reading the last sentence of this para. To explain a little more here. An illustration of an example or the final coding scheme would be most helpful.

Answer: We agree that the description of the purpose of coding was not sufficiently explicit. We have clarified this in the text by adding the following sentence: “Once extraction was completed, the information was coded to facilitate data synthesis and analysis”. Also, a complete description of the coding scheme is included in Supplementary Material 2 S2. We have now referenced this in the text.

Comment 12: Table 2 & 4: The subheadings inside the table could be italized for clarity.

Answer: Done.

Comment 13: Table 4: Was the categorization informed by a particular methodological paper? May be good to cite it or mention it here (Cresswell & Clark for example mentioned only later in the article).

Answer: Done. We have added citations in a footnote.

Comment 14: Same point highlighted in line 418-420 could explain why so little reports of mixed method studies are found in this review (commented in line 337). This may be added explicitly here to illustrate your point as a study limitation.

Answer: We agree with the comment. We have added the following sentence: “The search strategy used might have led to us to underestimate the true prevalence of articles reporting mixed methods studies in the field of palliative care and end-of-life research”.

Reviewer 2 Report

This study on "Characteristics and reporting quality of mixed methods articles in palliative care and end-of-life research: A methodological review" is a timely and well-written study that highlights the need for a more robust application and clearer write-up of mixed methods in the field of palliative care.

I have two small suggestions. The first one relates to Table 6. There are some formatting issues with the row numbers for Criteria indented on some but not in others. However, the main problem is in deciphering the results. The meaning of the Fisher Exact test significance remains unclear. Line 328-329 you report that "With respect to these two criteria, reporting quality was poorer in the studies using a convergent design than in those using sequential or multistage designs." In fact Criteria 3 also shows similar distribution (but not significant.) I am assuming that the contingency table is 5 X 3 (with 15 cells). In some cases, there are seven empty cells (Criteria 5). Although the advantage of the Fisher Exact test is the ability to cope with empty cells (in contrast to Chi-Square), the level of empty cells is still problematic. At least, you need to describe what the test is comparing. A sentence is required.

A small point that you highlight in line 420, where one of the limitations is that this study is solely based on English journals. Since most of these mixed-methods are written with a researcher from Europe, and therefore it seems that this methodology might be more popular in that region, a sentence on either the number of non-English palliative care journals that are published in Europe and/or an overview of the importance of mixed methods in Europe would be helpful to the reader.

Otherwise, a solid piece of work and very well discussed and written.

Author Response

Comment 1: However, the main problem is in deciphering the results. The meaning of the Fisher Exact test significance remains unclear. Line 328-329 you report that "With respect to these two criteria, reporting quality was poorer in the studies using a convergent design than in those using sequential or multistage designs." In fact Criteria 3 also shows similar distribution (but not significant.) I am assuming that the contingency table is 5 X 3 (with 15 cells). In some cases, there are seven empty cells (Criteria 5). Although the advantage of the Fisher Exact test is the ability to cope with empty cells (in contrast to Chi-Square), the level of empty cells is still problematic. At least, you need to describe what the test is comparing. A sentence is required.

Answer: Thank you for your insightful comment. First, the formatting issues were artifacts of the submission software since this information was correctly dispayed in our original Word file. We hope that the same error will not occurr. Second, we have clarified the rationale for using Fisher’s Exact Test in our study by adding the following sentence: “Since the sample size of articles included in the review was relatively small, and for several cells the expected count was less than 5, the standard Pearson’s Chi-Square Test was inappropriate. Therefore, Fisher’s Exact Test was used to determine the statistical significance of differences between groups”. Third, we have clarified the meaning of the significant relationships seen in Table 6 by adding the following sentences: “With respect to these two criteria, Fisher’s Exact Test failed to accept the null hypothesis that mixed methods research designs are equally likely to show the same reporting quality. Specifically, in both criteria the reporting quality was poorer in the studies using a convergent design than in those using sequential or multistage designs (i.e., the justification for using mixed methods and the integration process were both less frequently described in convergent mixed methods studies).”

Comment 2: A small point that you highlight in line 420, where one of the limitations is that this study is solely based on English journals. Since most of these mixed-methods are written with a researcher from Europe, and therefore it seems that this methodology might be more popular in that region, a sentence on either the number of non-English palliative care journals that are published in Europe and/or an overview of the importance of mixed methods in Europe would be helpful to the reader.

Answer: The findings indicate that mixed methods research is more or less equally used in Europe and North America, which is line with expectations. Therefore, we don’t think it is necessary to highlight this finding in the manuscript.